# Dietary Intake of the Italian PHIME Infant Cohort: How We Are Getting Diet Wrong from as Early as Infancy

**DOI:** 10.3390/nu13124430

**Published:** 2021-12-10

**Authors:** Federica Concina, Paola Pani, Claudia Carletti, Giulia Bravo, Alessandra Knowles, Maria Parpinel, Luca Ronfani, Fabio Barbone

**Affiliations:** 1Clinical Epidemiology and Public Health Research Unit, Institute for Maternal and Child Health—IRCCS Burlo Garofolo’, Via dell’Istria 65/1, 34137 Trieste, Italy; federica.concina@burlo.trieste.it (F.C.); claudiaveronica.carletti@burlo.trieste.it (C.C.); luca.ronfani@burlo.trieste.it (L.R.); 2Department of Medicine—DAME, University of Udine, Via Colugna 50, 33100 Udine, Italy; giulia.bravo@uniud.it (G.B.); maria.parpinel@uniud.it (M.P.); fabio.barbone@uniud.it (F.B.); 3Institute of Hygiene and Clinical Epidemiology, Azienda Sanitaria Universitaria Friuli Centrale, Via Colugna 50, 33100 Udine, Italy

**Keywords:** prospective cohort study, infants, energy macro and micronutrients intake, dietary reference value, food groups

## Abstract

Unhealthy dietary habits established in early infancy may lead to under or over nutrition later in life. This paper describes the energy, nutrients and food-type intake of 18-month-old infants belonging to the Italian PHIME cohort (*n =* 389) and evaluates adherence to the Italian dietary reference values (DRVs). Infant dietary data were collected using 7-day dietary records. Mean energy, macro and micronutrient intakes were estimated and compared with the DRVs. The percentage contribution of 19 selected food groups to total energy and macro- and micronutrient intake was determined with the aim of establishing the main food sources. Most infants’ diet shared common characteristics: poor variety, excessive intake of proteins (16.5 E% vs. 8–12 E% DRV) and saturated fatty acids (SFAs) (13.8 E% vs. <10 E% DRV), mainly derived from milk and dairy products, and low intake of total fats (33.2 E% vs. 35–40 E% DRV), polyunsaturated fatty acids (PUFAs) (3.1 E% vs. 5–10 E% DRV), vitamin D (1.1 vs. 15 µg/day DRV) and iron (4.5 vs. 8 mg/day DRV). The unbalanced distribution of macronutrients was reflected in energy intakes outside DRV ranges for more than half the infants. Public health interventions promoting healthy eating habits from early on, even from pregnancy, could yield significant short- and long-term health benefits.

## 1. Introduction

The first two years of life are a critical period of rapid physical growth and cognitive development and have an impact on short- and long-term health (e.g., infant neurodevelopmental disorders, childhood and adult obesity, adult cardiovascular diseases, adult diabetes, adult chronic diseases). This period of life is characterized by high energy and nutrients requirements and a rapid transition from a primarily milk-based diet (human milk and/or infant formula) to a varied diet with several food groups being consumed on a daily basis [1].

Poor dietary habits established as early as infancy, and even during fetal life, can lead to under or over nutrition [2]. The prevention of these conditions should start as early as the prenatal period with the promotion of healthy maternal behaviors during pregnancy, of breastfeeding, and of strong parental involvement in the shaping of their children’s dietary behaviors, starting from the transition to healthy foods during the complementary feeding period [3].

Chronic energy and macronutrient imbalances, derived from inadequate dietary intake, such as the consumption of nutrient-sparse calorie-dense food, and poor physical activity, contribute, together with genetic and socio-economic factors, to creating an obesogenic environment [4]. Indeed, obesity, in particular during childhood, is recognized as a major public health concern due to its significant (two- to threefold) increase in prevalence worldwide over the past few decades, especially in high-income countries [3]. Several longitudinal studies have described the strong association of pediatric obesity with persistence of obesity in adulthood, increased risk of future cardiovascular diseases and diabetes, as well as reduced life expectancy [5]. For this reason, a comprehensive dietary assessment among infants and toddlers, to monitor energy and nutrient requirements for growth and development, is essential to protect future health [1]. Furthermore, information on the diet and nutrition of young children allows for effective planning and implementation of national and international health programs [6]. To this end, a number of European countries, recognizing that infant diet is made of foods rather than individual nutrients, have translated dietary intake recommendations for infants and young children into food-based dietary guidelines to help provide caregivers with an indication of suitable age-appropriate foods to meet dietary needs and recommendations [7].

The main objective of the present paper is to provide a descriptive analysis of the dietary intake of energy, macro- and micronutrients in an Italian infant cohort at 18 months of age, while also assessing food group choices. The secondary objective of the paper is to evaluate the adherence of the cohort infants’ diet to the Italian dietary reference values (DRVs) [8].

Other studies have evaluated nutrient intake in Italian infant cohorts, observing that the majority of children in the age range 1–4 years old had a high intake of proteins, soluble carbohydrates, saturated fatty acids and sodium, and a low intake of iron, fiber, polyunsaturated fatty acids and vitamin D, compared to DRVs [9,10].

The added value of the present paper is its focus on food group choices, highlighting how a number of poor eating behaviors are already present from the early stages of life.

## 2. Materials and Methods

### 2.1. Study Population

The study population comprised 389 mother-child pairs belonging to a prospective cohort study conducted at the Institute of Maternal and Child Health IRCCS Burlo Garofolo in Trieste, Italy. The initial cohort was enrolled in 2007 as part of the ‘Public Health Impact of Long-Term, Low-Level Mixed Exposure in Susceptible Population Strata’ (PHIME) project. The PHIME project was funded by the European Commission’s Sixth Framework Programme for Research and Technological Development and the main aim of this prospective cohort study was to assess the association between low-level mercury exposure from food consumption during pregnancy and child neurodevelopment at the age of 18 months. The study protocol, with inclusion and exclusion criteria, is published elsewhere [11]. Only subjects who completed the 7-day dietary (7-DD) record (food diary) at 18 months of age were considered for this analysis [11,12].

### 2.2. Childhood Dietary Assessment and BMI Status at 18 Months of Age

Dietary data were collected using a food diary provided to mothers via e-mail or post, at hospital discharge with instructions on how to record type, quantity and method of feeding over a 24 h period on 7 days. The instructions also included a table with a list of common kitchen utensils that could be used as alternatives to measure solids and fluids (e.g., teaspoon, glass), and an indication of the estimated equivalent in grams. Moreover, mothers were also asked to report on their child’s state of health and on any medicine/supplements taken during the period of food diary completion. A telephone contact number was provided to mothers in case they had problems filling in the food diary.

Data extracted from the food diaries were analyzed using the Microdiet V2.8.6 software (Microdiet software—Downlee Systems Ltd., High, Peak, UK), containing the Italian Food Composition database for Epidemiological Studies in Italy—BDA [13], integrated with information from nutritional labels (e.g., commercial products, follow-on formula) and, in the case of breast milk, from literature [14,15]. Breast milk consumption was assessed on the bases of frequency of breastfeeding combined with estimated length of each feed [16]. Full details of the methodology are published elsewhere [17]. Demographic, education and social data of mothers and infants were obtained from the PHIME questionnaire [11]. All procedures were conducted by trained food technologists and nutritionists, who were fully familiar with brand names, composition of commercial products, food preparation methods and management of food composition data.

The nutritional analysis was performed on 27 food components: total proteins, carbohydrates (available and soluble, starch and fiber), lipids (total, saturated fatty acids SFA, monounsaturated fatty acids MUFA and polyunsaturated fatty acids PUFA; oleic acid, linoleic acid LA and linolenic acid ALA; cholesterol), minerals (sodium, potassium, calcium, iron, zinc) and vitamins (vitamin B1, vitamin B2, vitamin B6, vitamin B12, vitamin C, vitamin D, vitamin E expressed as a-tocopherol equivalent, niacin, folate). Energy (E) was calculated using the mean quantity intake of each macronutrient and applying the corresponding energy conversion factor.

Foods were classified into 19 food groups following the methodology proposed by Talamini et al., (2006) [18] and Sette et al., (2013) [19] and modified as follows: cereal-based products, milk and dairy products, eggs, vegetables, fruits, seeds and nuts, herbs and spices (also salt), fish, meat, fats and oils, beverages (water and beverage without sugar), sugars, sweets and desserts, sauces, tubers, baby foods and snacks, soups, beverages with added sugars (fruit juices, chamomile), pulses and cured meat.

The Body Mass Index BMI (kg/m^2^) was calculated (weight (kg)/height (m^2^)) based on the infants’ anthropometric data at 18 months, as measured by their community pediatrician during the health check and reported by mothers. These data were categorized according to the World Health Organization (WHO) Child Growth Standards [18] and expressed as BMI z score.

### 2.3. Ethics

The study was conducted according to the guidelines laid down in the Helsinki Declaration of 1975 and revised in 1983. The research protocol was approved by the ethics committees of the University of Udine and the Institute for Maternal and Child Health IRCCS Burlo Garofolo. All aspects of the study, including ethics, were monitored annually by the European Commission. All participating subjects gave their informed consent for inclusion before they were enrolled in the study.

### 2.4. Statistics

The general characteristics of the mother–child pairs are presented as frequency and percentage distribution for categorical variables and as mean, standard deviation (SD) and median for continuous variables. In order to verify comparability between the subjects who remained in the study and those who were lost at follow-up, a dropout analysis was conducted using the non-parametric signed-rank Wilcoxon test (for continuous variables) and two-tailed Fisher exact test (for categorical variables).

For each infant, the mean daily intake of energy, macro- and micronutrients, excluding the use of supplements, was calculated on a 7-day observation basis.

The mean, SD, median and interquartile range (IQR) were calculated for each daily energy and nutrient intake for all infants and by sex. Nutrient intakes were compared with the DRVs proposed by the Italian Society for Human Nutrition (SINU) [8] and expressed using different indexes: adequate intake (AI), reference intake range for macronutrients (RI), average requirement (AR), suggested dietary target (STD) and population reference intake (PRI). The percentage of infants with intake below, within or above the DRVs was estimated for each energy, macro- and micronutrient.

For each infant, the percentage contribution of the 19 food groups to the total intake of energy and of the separate nutrients, as well as to the total amount of food consumed, was estimated with the aim of establishing the main food sources.

The normality of the variables was tested using Kolmogorov-Smirnov test while the between sex differences were assessed using the Student t test or the corresponding non-parametric test (Wilcoxon, Mann–Whitney), depending on the distribution of the data. Statistical significance for all tests was set at a *p*-value of 0.05. SAS (version 9.4 SAS Institute INC., Cary, NC, USA) was used for all statistical analyses.

## 3. Results

In total, 900 mother–child pairs were enrolled in the cohort between 2007 and 2009, 632 (70.2%) of the infants remained in the cohort at 18 months, and 389 (43.2%) completed the food diary and are therefore considered in the present analysis. The characteristics of these mother–child pairs are shown in Table 1. The differences in sample size for some variables reported in the table are due to missing data in the questionnaires.

The dropout analysis carried out on all the variables presented in Table 1 showed a statistically significant difference in sex (*p* = 0.01) and birth weight (*p* = 0.002) of the infants. More specifically, the subjects considered in this analysis were mostly females (52.4% vs. 44.7%) and had a lower birth weight (3353.8, SD 460.9 g vs. 3454.3, SD 457.6 g), compared to the remaining children of this cohort.

The distribution of macronutrient intakes and the percentage of adherence to Italian DRVs [8] are shown in Table 2. The same data, stratified by sex, are reported in Online Appendix A.

The mean energy intake was 916.0 kcal/day (SD 196.1), with a statistically significant difference (*p*-value = 0.04) between females and males (892.5 kcal/day, SD 190.1 and 942.0 kcal/day, SD 199.9, respectively). Using as energy DRVs for the 1–2 years age range, 790–1050 kcal/day for females and 870–1130 kcal/day for males, 50.5% of females and 55.1% of males fell outside the Italian recommendation ranges. In particular, 30.9% of females and 37.3% of males had energy intake below the DRVs, and 19.6% of females and 17.8% of males above. The energy DRVs are calculated using the Schofield equation [20], which takes into account a median weight for each age extrapolated from Cacciari’s growth charts [21], and a median PAF of 1.39 for children under 3 years of age [8].

Overall, there is an evident imbalance in the energy contribution from macronutrients. In fact, while 95.4% of infants had excessive total protein intake when compared to DRVs, only 11.3% of infants fell within the RI for total fat. Moreover, a more detailed comparison in a subgroup of infants with complete anthropometric data at 18 months (*n* = 196), showed that, in all 196 infants, ‘actual’ protein intakes markedly exceeded estimated protein requirements: 3.3 g/kg/day (37.0 ± 10.1 g/day) vs. 1 g/kg/day (11.3 ± 1.4 g/day) for a mean body weight of 11.3 kg.

Similarly, we observed an imbalance in FA distribution, with the energy contribution of SFAs exceeding the DRV in 87.9% of infants, and only 5.1% of infants with PUFA intake in line with recommendations. In particular, LA and ALA DRVs were met by 4.9% and 29.6% of infants, respectively.

Seventy-three percent of infants met the DRV for total carbohydrate intake but approximately 93.3% exceeded the STD for soluble carbohydrates, while 42.4% did not reach the STD for fiber. Unfortunately, DRVs are not available for cholesterol, starch, and oleic acid. Statistically significant differences between females and males were observed for the intake of available carbohydrates, soluble carbohydrates, starch and fiber (all *p*-values < 0.05), with males presenting higher intake than females. (Appendix A).

The percentage deviation of micronutrient intakes from Italian DRVs [8], expressed using PRI and AI, are shown in Figure 1.

Setting the DRVs at 100%, the radar chart shows that the intake of B group vitamins was 1.5 to 2.5 times higher than recommended (mean intakes for Vitamin B1, B2, B6 and B12 were 0.6 mg/day SD 0.2, 1.0 mg/day SD 0.3, 1.0 mg/day SD 0.3 and 2.3 μg/day SD 1.1, respectively). Similarly, vitamin C intake significantly exceeded DRVs (mean intake 55.5 mg/day SD 34.8). On the other hand, the intake of vitamin D was less than one tenth of the recommended value (mean intake 1.1 µg/day SD 2.1) and iron intake was slightly over half the PRI (mean intake 4.5 mg/day SD 1.1).

A statistically significant difference between females and males was observed for the intake of vitamin B12 (females 2.4 ± 1.1 μg/day vs. males 2.2 ± 1.1 μg/day; *p*-value < 0.05).

Figure 2 presents the percentage distribution of the cohort population above and below the Italian DRVs. The same data, stratified by sex, are reported in Online Appendix A.

The majority of infants reported intake below Italian DRVs for the following micronutrients: 76.3% of infants for potassium, 69.7% for zinc, 87.7% for vitamin E, 62.7% for niacin and 72.8% for folate. None of the infants met the recommendation for vitamin D intake and only 3.1% of infants reached the PRI for iron. Contrarywise, the intake of calcium, sodium, and vitamin C were higher than recommended in 51.7%, 68.1% and 68.9% of infants, respectively. For the vitamin B group, over 85% of infants exceeded reference values.

At 18 months of age, sixty-five infants (16.7%) were still breastfeeding and twenty-one (5.3%) were formula feeding. All the formula used by infants was follow-on formula. Forty infants (10.3%) were still consuming baby foods (e.g., fruit and vegetable jars, baby cereals).

Figure 3 and Figure 4 report the percentage distributions of the six food groups with highest intake in total energy intake, quantity of food and macronutrients. For each variable, the category “others” includes all the remaining food groups.

Overall, our results show that, in our infants’ cohort, 33.6% of the energy intake came from milk and dairy products, the other main food sources being cereal-based products (20.9%) and baby foods and snacks (13.0%). Milk and dairy products were also the main contributors to total food intake (45.3%). The consumption of fish, eggs, dried fruits, and pulses was negligible (intake below 2%, included in the “Others” category in the Qty bar). Milk and dairy products also represented the main source of total proteins (41.8%) and soluble carbohydrates (40.3%) and the second food source of available carbohydrates (20.0%). Total proteins also came from meat (14.7%) and cereal-based products (13.9%) while available carbohydrates derived from cereal-based products (34.6%), baby foods and snacks (16.2%) and fruits (10.4%). Fruits accounted for 27.7% of fiber intake, and cereal-based products for 73.4% of starch.

The main source of total fats and FAs was the milk and dairy products food group. Other significant sources of cholesterol were meat (14.2%), eggs (12.3%) and baby foods and snacks (9.9%). Fats and oils and baby foods and snacks were the other two food groups providing fat intake, with PUFAs also deriving from cereal-based products (10.2%), meat (8.7%) and cured meat (8.7%).

Figure 5 and Figure 6 report the percentage distributions of the four food groups with highest intake in relation to total mineral and vitamin intakes. For each variable, the category “others” includes all the remaining food groups.

The principal contributors to B group vitamins and folate intake were milk and dairy products: 26.6% for vitamin B1, 62.6% for vitamin B2, 23.8% for vitamin B6, 55.5% for vitamin B12 and 24.9% for folate. Fish (23.9%) and meat (12.4%) were the main contributors of vitamin B12, with meat being also the main source of niacin (19.4%). The main source of vitamin D was fish, followed by milk and dairy products, with a percentage contribution of 35.2% and 21.6%, respectively. Concerning vitamin C, the principal food sources were fruits (41.5%) and vegetables (16.6%) while vitamin E mainly derived from fats and oils (27.7%), fruits (13.4%) and milk and dairy products (10.5%). The other two food groups that contributed to folate intake were vegetables (16.2%) and cereal-based products (15.9%). The principal source of all the minerals considered in this analysis, with the exception of iron, was the milk and dairy products food group with a percentage contribution of 32.0% for sodium, 38.9% for potassium, 72.2% for calcium and 40.7% for zinc. However, this food group was only the second source of iron (13.9%), in combination with baby foods and snacks (13.8%), while the main contributors were cereal-based products (18.2%). Other sources of sodium were cereal-based products and soups with a percentage contribution of 21.0% and 10.3%, respectively.

Anthropometric data were available only for 196 of the 389 infants with complete 7-day dietary records (50,4%), of these 97 (49.5%) were females and 99 (50.5%) males. BMI z-score distribution at 18 months of age for all infants and by sex, is reported in Table 3.

Fifteen infants (7.6%) were classified as overweight or obese, according to WHO standards (≥+2D) [22]. Of these, 10 were males (10.1%) and 5 were females (5.1%). Only one child, a female, was classified as underweight (≤−2SD) [22].

## 4. Discussion

The nutritional evaluation, at 18 months of age, of the dietary habits of 389 infants belonging to the Italian PHIME cohort, shows that their diet was characterized by poor variety, excessive intake of total proteins and SFAs, mainly derived from a single type of animal source, and a low intake of total fat, PUFAs, vitamin D and iron. This imbalance in the distribution of macronutrient intake was reflected in energy values below the Italian recommendation ranges for 34.1% of infants, and above for 18.7% [8]. It must be noted, however, that this percentage could be over or underestimated due to the inherent limitations of the methodology used to calculate energy intake, as reported in the Material and Methods section [8,20,21].

On the other hand, we have shown in a previous publication on this cohort, that similar imbalances also connoted the dietary habits of the mothers during pregnancy [23]. In particular, their diet was characterized by the excessive intake of soluble carbohydrates and SFAs. The adoption of healthy dietary behaviours during pregnancy and after childbirth is all the more important if we consider the impact of maternal diet on the development of the offspring’s food preferences in early life and on lifelong eating habits [2,24]. This is also confirmed by the results of our study on the diet of infants at 6, 9 and 12 months of age, which highlighted poor variety in food types and excessive intake of soluble carbohydrates, SFAs and total proteins, mainly deriving from animal-based products [25].

At 18 months of age, milk and dairy products were the main components of the infants’ diets, and were always the main sources of total proteins, available carbohydrates, SFAs, cholesterol and most of the minerals and vitamins. This was reflected in the fact that, in our cohort, all of these nutrients were found to be in excess of the Italian recommendations [8].

For almost all the infants in this study, total protein intake was above the RI (16.5 E% vs. 8–12 E%). This is in line with the protein intake levels described for Italian infant in other studies: 16.8 E% in the study conducted by Damianidi and colleagues [26] and 15 E% in the Nutrintake 636 study [9]. Because in our cohort most of the proteins derived from milk and dairy products, which contribute less than other animal sources (meat and fish) to iron intake and absorption, 96.9% of infants failed to reach iron PRI (mean iron intake around 50% of PRI). However, the percentage of infants below the reference value decreased to 37.5% when AR was used instead of PRI. The percentage of infants with inadequate PRI for iron in our cohort is higher than those reported in the other two Italian studies (71 and 79%) [9,10].

For most infants (73%) the intake of available carbohydrate was within the RI, while for 93.3% of infants, soluble carbohydrates exceeded the DRV, with only 20% deriving from fruit and over 30% from baby foods, snacks, sweets and desserts and sweetened beverages. A similar situation was also observed in the Nutrintake 636 study where 91% of infants at 18 months of age exceeded the STD for soluble carbohydrates [9].

More than half of the cohort infants (64%) had total fat intake below the RI, with an unbalanced distribution between the different types of FAs. Their diet was characterized by high SFA and low PUFA, resulting in below-recommended intake of essential FAs, which play a key role in infant neurodevelopment. These data are comparable to those from the Nutrintake 636 study that reports SFA intake above the RI for the 84% of infants at 18 months [9]. The main source of fats were milk and dairy products, while the contribution of fish and nuts was negligible (5% and 1%, respectively). This was likely due to the belief, common among parents and caregivers at the time of the establishment of the cohort, that these categories of food could elicit allergic reactions, as suggested by the then current international guidelines on timing of introduction of allergenic food [27].

A significant proportion of fats also derived from vegetable oils (mainly olive oil), commonly used in the Mediterranean area as dressing in many dishes and representing, for these infants, the main dietary contributor of PUFAs. Finally, despite national and international recommendations to limit the consumption of snacks and desserts for their high content in SFAs and added soluble carbohydrates, these are commonly used during infancy, as reported in other Italian studies [9,26] and also observed in this cohort (10.8%).

Concerning micronutrients, our results indicate that vitamin D, vitamin E, folate and iron intakes from dietary sources are insufficient, while sodium tends to exceed recommended levels. In particular, 100% of our infants had very low (7.3% of PRI) vitamin D intake. This seems to be a worldwide health problem since similar data are described in infants in the Netherlands [28], Spain [29], Finland [6] and Belgium [30]. The high prevalence of inadequate intake could be explained by the fact that this micronutrient is naturally found only in few foods (oily fish, egg yolk, and liver) and even food fortification (e.g., milk, breakfast cereals, margarine) has a modest effect on vitamin D status [31]. Fish was the main source of this vitamin in the infants of our cohort although, overall, the consumption of vitamin D rich foods was negligible. It is known that vitamin D is mainly synthesized by sunlight, but sun exposure is not always sufficient to offset low dietary intake, especially during infancy when skeletal growth rate is higher [8,31]. Recommendations on policies to improve vitamin D status in infants over 12 months of age, have been issued by national and international organizations. These include dietary guidelines, food fortification, vitamin D supplementation, and judicious sun exposure. This latter especially is strongly recommended in infants at higher risk of deficiency due to limited sun exposure (Nordic European population) or darker skin pigmentation [8,31].

The mean vitamin E intake was below the DRV in 87.7% of our infants. Low levels of vitamin E are common during infancy when stores are limited, and requirements are increased as a result of rapid growth. A recent worldwide review concluded that vitamin E intake was shown to be insufficient in 61% of the reviewed studies [32].

In our cohort, sodium intake exceeded the DRV by 25.7% in 68.1% of the infants, and derived mainly from milk and dairy products and cereal-based products. Similar results have been shown for infants in the Netherland [28], Spain [29] and Italy [9]. Since our study design did not include the measurement of urinary excretion, which is the only reliable method for estimating sodium intake, it is likely that the percentage excess we detected is in fact an underestimation. Although the evidence for a relation between sodium intake and blood pressure levels in infants is not conclusive, high sodium intake from an early age is likely to increase the threshold of the taste for salt and lead to excessive salt consumption later in life, thereby increasing the risk of cardio and cerebral-vascular diseases in adulthood [33].

The main strengths of this study were: (1) the adoption of a rigorous design (prospective cohort study) that allowed for accurate data collection; (2) the use of a 7-day dietary (7-DD) record, which is the gold standard for the assessment of dietary intake during infancy [34]; (3) the extraction of food composition and nutritional data using an opportunely integrated version of Microdiet V6.2.8. This software contains the Italian food composition database for epidemiological studies [13], compiled using standard methods established by the EuroFIR project (www.eurofir.org), and represents an important tool for epidemiological research, public health nutrition and education, clinical practice, and nutrition declaration of food labels.

The study has some limitations: (1) the loss to follow-up, however, no differences were observed in the main characteristics of mother-child pairs who remained in the study compared with those lost to follow-up. We cannot exclude that a certain degree of loss may have been due to the commitment required from parents to compile the 7-day dietary record; (2) when 7-DD records were incomplete and inaccurate, standard recipes and measures for data upload had to be adopted. However, this limitation is quite common in this type of studies; (3) the use of commercial products, which are generally characterized by poor nutritional labelling, may have limited the accuracy of our assessment of the intake of vitamins and minerals; (4) the methodology adopted to estimate the consumption of human milk (frequency of feeds, perceived length of each feed by mothers) and the use of a literature-derived composition, do not take into consideration inter- and intrasubject variability and may therefore contribute to the inaccuracy of the assessment of nutrients intake [17]. This methodology, however, has been used before in other infant cohort studies [35,36] and the percentage of infants in our cohort that were still breastfed at 18 months was relatively low (16.7%); (5) the data were collected over ten years ago and dietary habits in infants may have changed since then. However, the main nutritional concerns emerging from this assessment, in particular inadequate macro- and micronutrient intake, and the presence of poor dietary habits early on in life, continue to be relevant and reflect those which are still recognized as the most problematic aspects of nutrition in infancy [9,10,27]. As a final consideration, it is interesting to note that the poor dietary behaviours we observed in the infants of our cohort reflect closely those of the pregnant women belonging to the same population [23], and of the adult population in general [37].

## 5. Conclusions

The nutritional assessment of the diet of the infants of the Italian PHIME cohort at 18 months of age, highlights a number of inadequate dietary behaviours: the low consumption of fruits and vegetables, fish and pulses, typical of the Mediterranean diet and recommended by the Italian Food-based Dietary Guidelines [38], the predominant consumption of milk and dairy products as sources of proteins and fats, and the protracted use of baby foods and follow-on formula as late as 18 months of age. Thus, the promotion of healthy eating habits starting at a very early age, in fact as early as the time of introduction of complementary food, could yield substantial short- and long-term health benefits.

## Figures and Tables

**Figure 1 nutrients-13-04430-f001:**
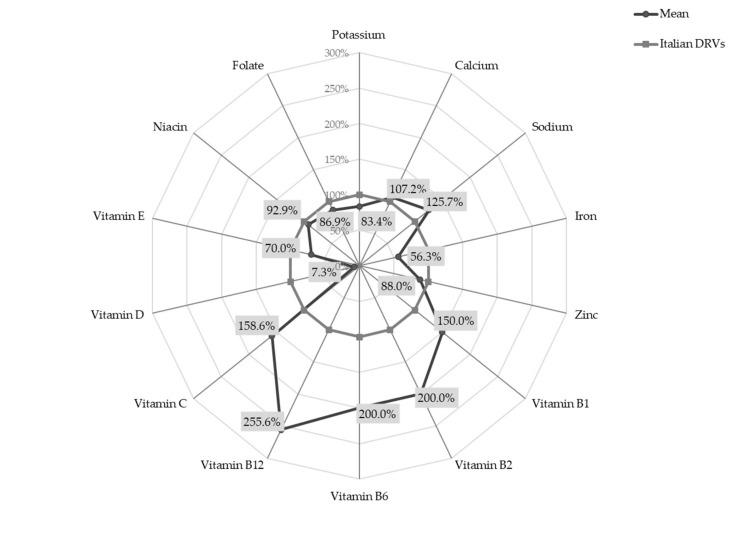
Micronutrient intakes in infants at 18 months of age, expressed as percentage deviation from Italian DRVs (PRI and AI) (*n* = 389). Abbreviation: DRVs, dietary reference values.

**Figure 2 nutrients-13-04430-f002:**
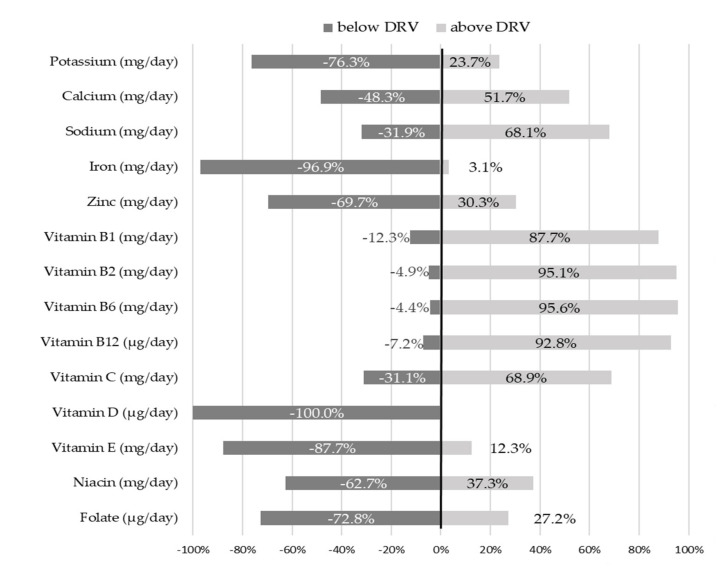
Adherence of the cohort population to Italian micronutrient DRVs (*n* = 389). Abbreviation: DRV, dietary reference value.

**Figure 3 nutrients-13-04430-f003:**
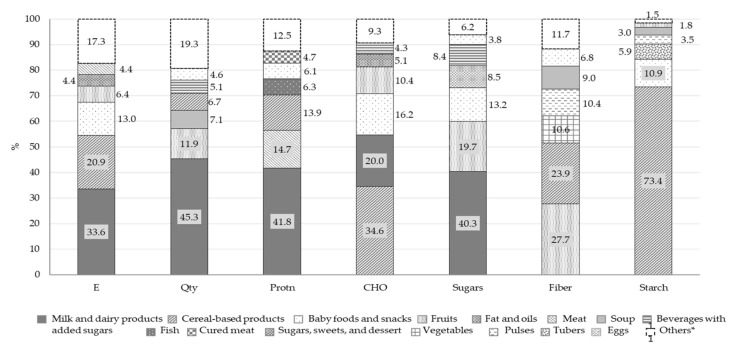
Percentage contribution of selected food groups to total intake of energy, food (Qty), proteins, available carbohydrates, sugars, fiber and starch (*n* = 389). * For each variable, the category “Others” includes all the food groups not considered independently. Abbreviations: E, energy; Qty, quantity of food; Protn, total protein; CHO, available carbohydrates.

**Figure 4 nutrients-13-04430-f004:**
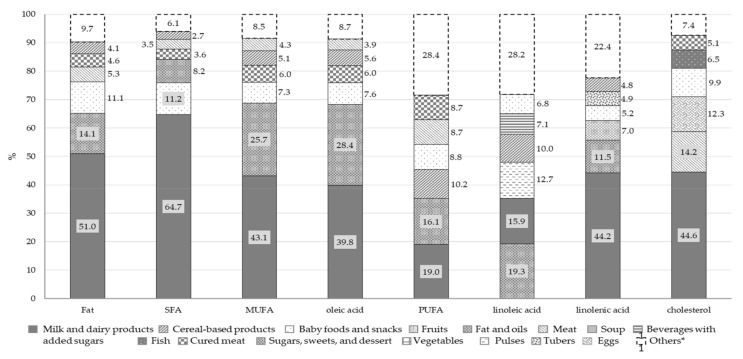
Percentage contribution of selected food groups to total intake of fats (*n* = 389). * For each variable, the category “Others” includes all the food groups not considered independently. Abbreviation: SFA, saturated fatty acids; MUFA, monounsaturated fatty acids; PUFA, polyunsaturated fatty acids.

**Figure 5 nutrients-13-04430-f005:**
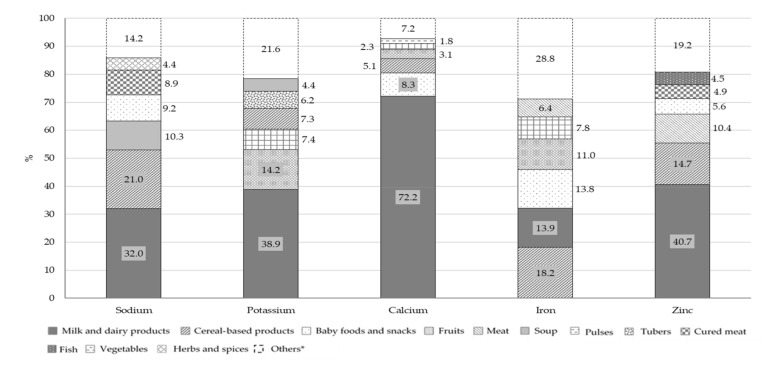
Percentage contribution of selected food groups to total intake of minerals (*n* = 389). * For each variable, the category “Others” includes all the food groups not considered independently.

**Figure 6 nutrients-13-04430-f006:**
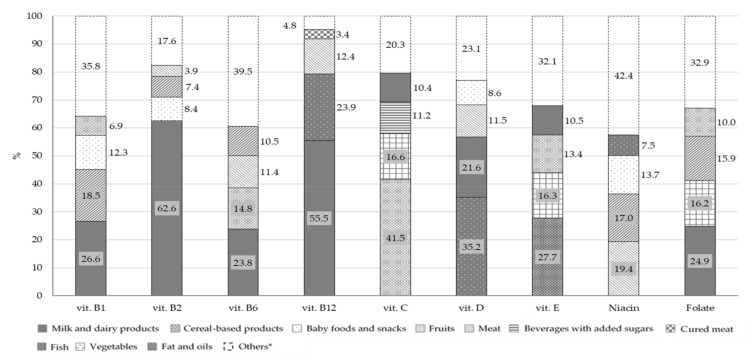
Percentage contribution of selected food groups to total intake of vitamins (*n* = 389). * For each variable, the category “Others” includes all the food groups not considered independently.

**Table 1 nutrients-13-04430-t001:** General characteristics of mothers at enrollment and children at birth (*n* = 389).

Characteristic	*n*	%
Maternal Nationality		
Italian	352	90.5
Foreign	27	6.9
Not reported	10	2.6
Maternal Marital Status		
Married/living with partner	347	89.2
Separated/divorced	11	2.8
Single/not living with partner	21	5.4
Not reported	10	2.6
Maternal Education		
Completed primary school	3	0.8
Completed secondary school	54	13.9
Completed high school or equivalent	180	46.3
Bachelor degree or higher	143	36.7
Not reported	9	2.3
Sex of Child		
Male	185	47.6
Female	204	52.4
**Characteristics**	**Mean**	**SD**
Maternal Age (years; *n* = 374)	33.8	4.2
Birth Weight of Child (grams; *n* = 386)	3353.8	460.9
Birth Length of Child (cm; *n* = 385)	50.0	2.2

Abbreviations: *n*, number of subjects; %, percentage of subjects and SD, standard deviation.

**Table 2 nutrients-13-04430-t002:** Distribution of macronutrients intake in infants at 18 months of age (*n* = 389).

Macronutrients	Mean ± SD	Median (IQR)	DRV	% of Infants Below the DRV	% of Infants within the DRV	% of Infants above the DRV
Total protein (g/day)	37.8 ± 10.0	36.9 (31.3–44.3)				
Total protein (E%)	16.5		8–12 E% ^1^	-	4.6	95.4
Total fat (g/day)	33.9 ± 9.4	32.7 (27.2–39.8)				
Total fat (E%)	33.2		35–40 E% ^1^	64.0	11.3	24.7
Saturated fatty acids (g/day)	14.1 ± 4.6	13.5 (10.7–17.2)				
Saturated fatty acids (E%)	13.8		<10 E% ^2^	12.1	-	87.9
Monounsaturated fatty acids (g/day)	11.6 ± 4.0	11.1 (8.8–13.9)				
Monounsaturated fatty acids (E%)	11.4		10–15 E% ^1^	33.9	56.1	10.0
Oleic acid ^3^ (g/day)	10.2 ± 3.9	9.8 (7.4–12.4)				
Polyunsaturated fatty acids (g/day)	3.2 ± 1.3	3.0 (2.3–3.8)				
Polyunsaturated fatty acids (E%)	3.1		5–10 E% ^1^	94.9	5.1	-
Linoleic acid (g/day)	2.4 ± 1.1	2.2 (1.7–2.8)				
Linoleic acid (E%)	2.3		4–8 E% ^2^	95.1	4.9	-
Alfa-linolenic acid (g/day)	0.4 ± 0.1	0.4 (0.4–0.5)				
Alfa-linolenic acid (E%)	0.4		0.5–2.0 E% ^2^	70.4	29.6	-
Cholesterol ^3^ (mg/day)	119.7 ± 42.6	118.9 (90.8–143.7)				
Available carbohydrates (g/day)	122.6 ± 30.1	120.9 (101.7–139.1)				
Available carbohydrates (E%)	53.5		45–60 E% ^1^	11.1	73.0	15.9
Soluble carbohydrates (g/day)	53.8 ± 16.8	51.5 (42.2–62.6)				
Soluble carbohydrates (E%)	22.0		<15 E% ^2^	6.7	-	93.3
Starch ^3^ (g/day)	56.0 ± 22.7	54.5 (40.1–70.4)				
Fiber (g/day)	7.3 ± 2.9	7.0 (5.2–9.2)	8.4 g/1000 kcal ^4^	42.4	-	57.6

^1^ Below DRV: extreme of the range not included (<); within DRV: extremes included above DRV: extreme not included (>). ^2^ Below DRV: extreme included (≤); above DRV: extreme not included (>). ^3^ DRVs were not available. ^4^ Below DRV: extreme not included (<); above DRV: extreme included (≥). Abbreviation: SD, standard deviation; IQR, InterQuartile Range; DRV, dietary reference Value; E%, energy percentage.

**Table 3 nutrients-13-04430-t003:** BMI z-score distribution in all infants and by sex (*n* all infants 196; *n* females 97; *n* males 99).

	Females *n* (%)	Males *n* (%)	All Infants *n* (%)
−2SD	1 (1.0)	-	1 (0.5)
−1SD	5 (5.2)	7 (7.1)	12 (6.1)
0SD	65 (67.0)	63 (63.6)	128 (65.1)
+1SD	21 (21.7)	19 (19.2)	40 (20.4)
+2SD	5 (5.1)	9 (9.1)	14 (7.1)
+3SD	-	1 (1.0)	1 (0.5)

Abbreviations: *n*, number of subjects; %, percentage of subjects; SD, standard deviation.

## Data Availability

The data described in the manuscript, in the code book and in the analytical code will not be made available because we do not have an accessible repository in which to deposit them.

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
