# Peer review of "Dietary Intake of the Italian PHIME Infant Cohort: How We Are Getting Diet Wrong from as Early as Infancy"

_nutrients, 2021, doi:10.3390/nu13124430_

Round 1

Reviewer 1 Report

GENERAL COMMENTS
This is a potentially very important paper on an interesting topic. However, several limitations dampen enthusiasm.
The Abstract is absent any quantitative result. The authors go straight from describing the methods to discussing results without ever providing any quantitative results. There also is evidence that dietary interventions in pregnancy can affect fetuses in utero. What protections against bias were taking? There are steep reductions in number of subjects available for analyses. This is particularly true for dietary and anthropometric data. What might be the effect of selection bias? The food groups are very heterogeneous. Wouldn’t it be more sensible to use or devise a summary measure of overall dietary quality. On the other hand, however, much of the discussion focuses specifically on nutrient excesses and deficiencies. So, this is not particularly damning. It would, however, be interesting to look at the overall pro-inflammatory effect of the diet at some future point (not necessarily in these analyses.
Overall, it seems that milk and dairy products are driving most of the specific food group and nutrient intakes evident here. What implications does this have for clinical medicine and public health? What about the role of dietitians in working with mothers to change behavior?
SPECIFIC COMMENTS
Abstract
Lines 17-8. Wording is awkward. Move “at 18 months of age” up to read “intake of infants at 18 months of age” as this refers to the age of the children not the cohort.
Line 19. Replace “by” with “using”
Lines 21-2. Sentence is garbled. Is the word “in” before “total intake of energy” correct? Shouldn’t it read “and intake of total energy”? The next sentence also is poorly written “infants shared a low-varied diet”?
Lines 22-6. Results are discussed without ever providing any quantitative data in support of the statements provided
Introduction
Lines 38-9. Are infant formulas always milk-based?
Line 41. Some would argue that dietary habits of the mother can condition responses in her fetus. So, the argument is well taken. However, it can be targeted back towards even earlier periods in development.
Line 42. The wording “already during” is awkward and incorrect
Line 52. While the focus on macronutrients is not unreasonable, there is much more to inadequate dietary intake than macronutrients of total caloric intake. Indeed, one of the sequelae of transitioning to a healthier diet includes a displacement of nutrient-sparse, calorie-dense food with nutrient-dense, calorie-sparse food.
Lines 59-61. If the focus is on a comprehensive dietary assessment among infants and toddlers then, as the authors say, specific nutrient requirements should be taken into account.
Lines 64-8. I really like the focus on food-based dietary guidelines. The goal should be to focus on nutrient-dense foods.
Line 70. Reword “energy, macronutrient, and micronutrient intake”
Materials and Methods
Line 79. Are these singlet pairs? In other words, are twins and other multiple births excluded? Also, did the authors take into account birth order? It is well-known that birth order and spacing can affect a wide variety of health outcomes.
Lines 87-8. By including only subjects who completed the 7-Day dietary record was selection bias introduced?
Line 94. Fix wording “weighting to measure”
Lines 122-7. The 19 food groups appear to be quite heterogeneous. Some are whole categories of whole foods while others include ultra-processed foods (i.e., sweets and desserts, sauces, baby foods soups). What are cured meats? Are these highly processed foods? Would it be better to simply categorize food as to whether they are whole foods eaten as such, whole foods incorporated into homemade recipes, and highly processed foods?
Line 128. The BMI is an inappropriate measure of late adjusted for height in young children. There are better measures to describe relative weight in children. Mention at the outset that they used BMIz (stratified by age). They also may want to try age-specific weight-for-height standards. One also could just look at the relationship between weight and height independent of age, as we did in war zones in Africa.
Lines 130-2. It seems as though the authors are anticipating this problem. However, why were data available only for hundred 96 infants. Also, I thought that there were 900 mother child pairs. How could 196 be 50.4% of the total? In the results section this is revealed. These are only individuals who are eligible because the food diaries were completed. What kind of selection bias does this introduce?
Lines 147-8. Why were supplements excluded? Were separate analyses conducted with supplements included?
Lines 156-7. Is the wording “in the total intake” correct? Shouldn’t it be “and the intake of total energy”
Results
Lines 165-7. Why was there such a large decline in the size of the cohort?
Line 187. Analyses were stratified, not divided, by sex.
Lines 193-5. Shouldn’t energy and macronutrient intake be described per unit body weight? Conflict is a lot of potential variability here.
Table 1. These macronutrient results are interesting and salient one should be highlighted in the Abstract.
Figure 1. These micronutrient results also are very interesting and could be highlighted in the Abstract of the paper. Vitamin D deficiencies seem to be rampant, as they are in other populations.
Figure 2 is very helpful. It really would be interesting to highlight some of these results.
Line 256. What is follow-on formula?
Figure 3. Check title.
Figure 4. Check title.
Figure 5. Check title.
Figure 6. Check title.
Table S1. P-values should go to 2 places unless there are leading zeroes.
Discussion
Line 348. This “imbalance”
Line 355. These “imbalances”
Lines 358-9. Somewhere around here mention should be made of the importance of maternal diet even before childbirth. There is a nice literature in support of conditioned responses at very early ages.
Line 378. The authors should call out” soluble carbohydrates” by mentioning that these are primarily sugar-sweetened soft drinks and other highly processed foods. Of course, this is consistent with other results reported.
Lines 382-5. What about intakes of polyunsaturated specific fatty acids? The authors stated that much vitamin D is coming from fish. This also would be a good source of omega-3 fatty acids.
Lines 436-51. Selection biases should be included among limitations cited.

Reviewer 2 Report

This paper report dietary intake of 18 m old children in Italy. Data are based on dietary data of 389 children from a prospective cohort study including 900 mother-child pairs from the 79 Institute of Maternal and Child Health IRCCS Burlo Garofolo in Trieste, Italy. Out of the 389 children anthropometrics of 196 children were available and included in the analysis. The objective was to assess and report intake of energy and nutrients as well as food groups.

Dietary intakes were assessed using a 7 d food records and described as energy- and nutrient intakes in total and in compared with Italian guidelines proposed by the Italian Society for Human Nutrition.

The paper includes a lot of interesting data and could be divided into two papers.

Title

The title does not fit the paper, it is mostly recommended to start with a key word. I suggest you to start with; Dietary intake in young Italian children

Introduction

In line 37: This period of life is characterized by higher energy and nutrients requirements .. Please change from higher energy. .. to high energy or include compared to (define to what you compare with)…..?

Line 42-47 and maybe elsewhere: I agree with the text but you do not study maternal health or its association with her infant/child in the present paper.

Line 52-59 public obesity and other publish health diseases are described while there is poor data on what is known of diet or nutrient intakes in young children.  Italian guidelines of dietary intakes in infants and children is not reported here. Moreover, how are child care in 18-m old Italian children provided, do they stay at home or go to preschool? This is important to know and may be a reason for the high drop-outs

Line 74-76: You write that other have evaluated nutrient intake in Italian infant cohorts and I suggest you to make a better description of their findings, what is known in Italy and  maybe, also international reports including energy, protein and other nutrients of interest related to your study before your Objectiv.

You can also include something about healthy diet and what amount children should/need to consume of certain food groups as milk, fruit, vegetables and more, and then be able to compare portion sizes with some guidelines.  ( Food and Agriculture Organization of the United Nations and World Health Organization. Sustainable healthy diets: guiding principles. 2019. Internet: http://www.fao.org/3/ca6640en/ca6640en.pdf and Willett W, Rockström J, Loken B, Springmann M, Lang T et al. Food in the Anthropocene: the EAT–Lancet Commission on healthy diets from sustainable food systems. Lancet. 2019;393:447–92.)                                       

 Materials and Methods

First: It will be better to add and start the number of children included in this paper/analysis than start with reporting 900 mother and child pairs

Next: Dietary assessments and calculations; Who were making the dietary calculations? Please provide information about nutrients that were missing in the data base and not could give sufficient information of intake and how you did manage this.

Food groups: Is potatoes included in vegetables or why are the not reported?

Statistics

Explain way (or when) you are using both mean and median and other methods to describe distributions as in table 2?

Results

Data should not be double reported in tables and in text. From line 176 and further data results are doubled, please check this carefully and rewrite.  Line 193, also double reported from suppl. table. In the supplemental table, energy intakes are reported both as kcal and KJ, why?

Line 181-185: Please rewrite and include the p value in the same sentence as the differences.

In line 284 -287: “Moreover, a more detailed comparison in a subgroup of infants with complete anthropometric data at 18 months (n 196), showed that the ‘real’ protein intakes markedly exceeded the estimated protein requirements for all 196 infants: 37.0 g/day (SD 10.1) vs. 11.3 g/day (SD 1.4).” That is a recalculation from g/kg and my suggestion is that you report the intake of protein as g/kg and compare it with 1 g/kg.  It will make it possibly to compare with other studies worldwide in your discussion.

Table 1: could be improved

Table 2: I suggest that you only include intake data from nutrients with sure/sufficient data and with a DRV-value to compare with. Maybe cholesterol and starch are such as well as your data on fatty acids as alfa - linolenic acid and others, are they sufficient?  Do you need to report both mean and median intake % of infants as well as % of infants below the DRV; % of infants within the DRV and above the DRV?  Explain this in statics.

Table 3

Only 7.6 % of the children is reported to be obese/overweight, a low number compared to other studies but not further analyzed or discussed in this paper, why?  It does not match  lines 52-59.

Figure 1 and 2

I suggest a table with mean or median intake should be used together with DRV for Italian child making It possibly to compare data across studies.

Figure 3 and 4

I suggest you change and present data in a table similar as in the paper in your ref no 19 by Sette et al. The table in ref no 19 gives a better overview and is easier to read.

In the text after the figures, data is doubled, please make a change also here.

Discussion

From line 345 and further you should compare your result better with others for example how much protein (g/kg), fatty acids, Iron/zink, breastfeeding and other nutrients is reported and if you want to discuss health outcomes  from those studies it can be added.

In line 399- 400: In particular, 100% of our infants had very low (7.3% of PRI) vitamin D  intake.” This deficiency seems to be a worldwide” here you use deficiency but that was not measured.  Please rewrite – the low intake is not deficiency and if you want to discuss this please use data that combine vitamin D intake and measured vitamin D status. There are several paper published regarding Vitamin D intake and Serum-25 (OH)D in young children after the ESPGHAN 2013.

Line 403: please note that the ESPGHAN Committee on Nutrition recommends that national authorities adopt policies aimed at improving vitamin D status using measures such as dietary recommendations, food fortification, vitamin D supplementation, and judicious sun exposure, depending on local circumstances.

Conclusion

Line 460-471 is more like a discussion and must be deleted from conclusions. Moreover, associations between mother and child was not analyzed and is therefore not appropriate to discuss and reported in this paper although that would be very interesting.
